# Material Properties, Characterization, and Application of Microcellular Injection-Molded Polypropylene Reinforced with Oyster Shells for Pb(II) Adsorption Kinetics from Aqueous Solutions

**DOI:** 10.3390/polym18010110

**Published:** 2025-12-30

**Authors:** Minyuan Chien, Naveen Bunekar, Cabangani Donga, Pontsho Mbule, Tlou Nathaniel Moja, Shyhshin Hwang

**Affiliations:** 1Department of Vehicle Engineering, Chien-Hsin University of Science and Technology, Taoyuan 320678, Taiwan; jackyaren@uch.edu.tw; 2Department of Chemistry, Chung Yuan Christian University, Taoyuan 32023, Taiwan; naveen.bunekar@gmail.com; 3Department of Physics, College of Science, Engineering and Technology, University of South Africa, Florida Campus, Johannesburg 1709, South Africa; dongac@unisa.ac.za (C.D.); mbuleps1@unisa.ac.za (P.M.); 4Institute for Nanotechnology and Water Sustainability, College of Science, Engineering and Technology, University of South Africa, Florida Campus, Johannesburg 1709, South Africa; 5Department of Mechanical Engineering, Chien-Hsin University of Science and Technology, Taoyuan 320678, Taiwan

**Keywords:** oyster shell nano powder, polypropylene, microcellular injection molding, compounder, Lead Pb(II), adsorption

## Abstract

Microcellular injection-molded polypropylene/oyster shell nano-powder (PP/OSP) composites show potential as adsorbent materials for reducing toxic metal ion contamination in groundwater. This study investigates the material properties of PP/OSP composites and evaluates their Pb(II) adsorption performance in aqueous media. The effects of key operational parameters, including contact time, pH, and initial Pb(II) concentration, were examined to determine the optimal conditions for heavy metal remediation. The composites were characterized using XRD, SEM, FTIR, and TGA to assess their crystalline structure, surface morphology, functional groups, and thermal stability, respectively. Adsorption isotherm analysis indicated that the Pb(II) uptake behavior followed both the Freundlich and Temkin models. Kinetic studies showed that the adsorption process was best described by the pseudo-first-order model. The maximum adsorption capacity for Pb(II) removal was determined to be 13.89 mg/g.

## 1. Introduction

Oyster is a mollusk and a marine shellfish, also known as “milk of the sea.” It is a major marine resource with high nutritional value. Recently, researchers have focused on natural fillers, such as calcium oxide (CaO), which have a spherical shape. OSP is an inexpensive and readily available resource that can be used in thermal stabilizers, heavy metal adsorption, and flame-retardant applications [1]. Furthermore, Taiwan is an island with an abundance of marine resources, including oysters, abalones, and clams. One practice in a circular economy is reusing such materials to produce new products, among which their use as polymer fillers is the most effective application. However, microplastics pose a potential risk to human health through transmission and bioaccumulation in the food chain [1] as they have already been widely detected in food [2,3]. In the future, the environmental risks associated with microplastic exposure are likely to increase, as global plastic production is projected to reach 318 million tons per year by 2050 [4]. Moreover, due to their small particle size, large specific surface area, and hydrophobic characteristics, microplastics can act as carriers that transport heavy metals from rivers to marine environments [5]. Heavy metals such as Cu, Cr, and Pb—commonly detected pollutants in various ecosystems [6]—have frequently been found adsorbed onto environmental microplastics [7], often at concentrations comparable to or even higher than those present in the surrounding sediments [8].

Although microplastics are capable of adsorbing heavy metals [8,9], their adsorption capacities differ depending on both the type of microplastic and the specific metal involved, largely due to variations in physicochemical properties. For instance, polystyrene and film-type microplastics exhibit a higher adsorption capacity for Cu(II) compared with polyvinyl chloride, polyethylene, fishing line fibers, and bottle-cap particles, primarily because film microplastics possess more favorable surface characteristics [10]. Among the common heavy metals, Pb shows a stronger affinity for polypropylene (PP) than Cu or Cd, as it more readily interacts with functional groups present on microplastic surfaces, thereby enhancing adsorption [11]. Particle size also plays a critical role in determining adsorption behavior. In natural environments, microplastics undergo fragmentation into smaller particles due to physical and chemical weathering processes [12], which subsequently influences their capacity to adsorb heavy metals [13].

Moreover, polypropylene (PP) is one of the most widely used polymers in packaging and industrial applications due to its low cost, low density, and excellent chemical resistance [14]. It has also been extensively utilized in biomedical materials because of its non-cytotoxic nature, ease of processing, and recyclability [15]. However, PP inherently lacks antibacterial properties, making it susceptible to bacterial colonization [16]. Therefore, surface modification is often required to enhance its antibacterial activity or improve its biocompatibility. Incorporating antimicrobial agents into PP, however, is challenging because of its hydrophobic surface and limited functional groups [17]. To address these limitations, various surface modification techniques, including chemical treatment, flame treatment, Co-γ irradiation, corona discharge, ultraviolet exposure, and plasma treatment, have been applied to improve PP’s surface characteristics [18].

Several studies have reported modifying the surface of PP by incorporating various fillers, including SiO_2_ [19], CaCO_3_ [20], Al_2_O_3_ [21], nano-clay [22], and ZnO nanoparticles [23]. The specific applications of these materials depend on the physicochemical properties of the inorganic nanoparticles used. Joudeh et al. [24,25] modified PP by first exposing it to γ-irradiation in the presence of acetylene and then blending it with Ag nanoparticles through extrusion, resulting in nanocomposites with notable antimicrobial activity. Other studies have demonstrated that ZnO is an effective antibacterial additive for PP [26]. In contrast, the present study explores the incorporation of nano oyster shell powder (OSP) as a filler to enhance the mechanical, chemical, and physical properties of PP, as well as to improve its adsorption efficiency [27].

Several studies have explored the recycling of OSP waste and its conversion into value-added products [27,28]. Hsu [29] reported that oyster shell powder (OSP) can effectively adsorb certain heavy metals, including Pb(II) and Ni(II), from wastewater. OSP can also be processed using hydrothermal methods, and related technologies have been applied to fabricate porous scaffolds [30] and to produce raw materials for bone tissue engineering. As a recyclable material, oyster shell offers strong potential for adsorbing heavy metal ions from wastewater and contaminated groundwater [31]. Reusing oyster shells as an alternative, low-cost, and non-toxic adsorbent not only reduces environmental impact but also provides economic and practical value by converting waste into a useful resource [32]. Accordingly, reusing waste oyster shells as an adsorption material for wastewater remediation provides an innovative approach to environmental improvement [33]. It also increases the value of waste and contributes to the development of the oyster meat industry in South Africa. The main objective of this study was to investigate the adsorption potential of polypropylene/oyster shell powder as an effective adsorbent for metallic ion uptake in solution and to characterize the polypropylene/oyster shell powder adsorption mechanisms [34,35]. Batch adsorption experiments were conducted to study the effects of contact time, pH value, initial concentration, and dosage on adsorption efficiency.

## 2. Materials and Methods

### 2.1. Materials

Nano OSP with an average particle size of 100 nm and light gray color was supplied as received by Hans Global Textile Co., Ltd., Tainan, Taiwan, while Polypropylene (PP) with part number K1035 and a melt flow index of 38 g/10 min was provided by Taiwan Formosa Co., Ltd., Kaoshiung, Taiwan. The chemical composition of PP is (C_3_H_6_)n, and OSP consists of 98.2% CaO. In addition, 99% grade Pb(NO_3_)_2_ was supplied by Merck KGaA, Darmstadt, Germany.

### 2.2. Foamed Injection Molding Machine and Mold

Microcellular injection molding was performed using a 100-ton Arburg 420C injection molding machine equipped with MuCell^®^ capability from Zhongli, Taoyuan, Taiwan. Nitrogen (N_2_) served as the physical blowing agent. A supercritical fluid (SCF) booster can pump nitrogen gas up to 34 MPa, and SCF is injected into the polymer melt. A shutoff nozzle is used to maintain the polymer melt and scN_2_ as a single phase at around 10 MPa in the barrel. Upon injection, thermal instability occurs, causing cell nucleation and growth during the injection process. In turn, foamed polymer is formed (Figure 1). Processing parameters are listed in Table 1. PP/OSP nanocomposites were fabricated through the following steps. The OSP powder was obtained as received. Then, we sprayed dispersed wax onto the PP pellets and added OSP powder to them. Mixing was completed by shaking the OSP powder on the surface of the PP pellets inside a plastic bag. The mixing ratio for PP/OSP bio-composites is shown on Table 2.

The nano-OSP content was 2, 5, 7, and 12 wt%, respectively, and the mixing ratios are shown in Table 1. The mixing process was performed by spraying liquid wax (1 vol.% of the batch weight, 600 g, 6 mL for 7 wt% OSP) onto the surface of the PP pellets inside a plastic bag, and the powder dispersion was completed by hand-shaking the plastic bag. One batch of 600 g OSP/PP composites produced approximately 25 molds of ASTM D638 tensile specimens. Microcellular injection molding (as shown in Figure 1) was conducted using an Arburg-420C 1000 injection molding machine equipped with Mucell^®^ capability. Foamed samples were crushed using a plastic grinder for the adsorption study. The crushed samples size is around 4 mm.

The thermal decomposition temperature was determined by thermogravimetric analysis (TGA), which was performed using an SII TG/DTA6200 instrument with approximately 10 mg of sample. The measurements were carried out in air from 40 to 600 °C at a heating rate of 10 °C/min. The cellular morphology of the foamed composites was analyzed using a JEOL JSM-6360 scanning electron microscope (SEM). The PVT property was measured using Smart RHEO of CEAST, Pianezza (TO), Italy. X-ray diffraction (XRD) analysis was carried out using a PANalytical PW3040/60 X’Pert Pro diffractometer with Cu Kα radiation (45 kV, 40 mA) and a wavelength of λ = 1.54 Å. The diffraction patterns were recorded over a 2θ range of 2–70° at a scan rate of 3°/min for the polymer composite samples.

## 3. Kinetic and Equilibrium Sorption Studies

The study employed a systematic batch adsorption methodology to investigate the effects of pH, initial metal concentration, and contact time on Pb(II) removal. For each experiment, approximately 0.17 g of the PP/OSP was measured and reacted with aqueous solutions of the metal ions [36]. Lead nitrate was dissolved in deionized water to prepare a 100 mg/L of Pb(II) stock solution. The effects of pH, contact time, initial concentration, and PP/OSP dosage on adsorption were examined through batch adsorption experiments. Initial concentrations ranging from 0.10 to 1.25 mg/L were prepared from the 100 mg/L stock solutions.

These mixtures were agitated on a mechanical shaker at a constant speed of 150 rpm and maintained at a controlled temperature of 25 °C. Subsequently, the pH of the aqueous solutions was carefully adjusted from 2 to 12 using 0.1 M HCl and NaOH. At the same time, other parameters were held constant to determine the influence of the pH on metal ion adsorption. The effect of the initial metal ion concentration was investigated by exposing the PP/OSP to solutions containing 0.1 to 1.25 mg·L^−1^ of Pb(II). Furthermore, the effect of contact time was examined by measuring the adsorbed metal ions at predefined intervals ranging from 5 to 30 min at 5 min intervals.

After the experiments, the mixtures were filtered, and the metal ion concentrations in the supernatants were quantified using ICP-OES, ensuring the reliability and accuracy of the results. The quantities of adsorbed metal species at equilibrium (Q_e_) and their removal efficiencies (R) were calculated using Equations (1) and (2), expressed as follows [37]:(1)$$Q_{e}\;=\;(\frac{Co-Ce}{v})m$$
(2)$$\%R=(\frac{Co-Ce}{Co})100$$

Here, Q_e_ represents the amount of adsorbed metal species at equilibrium; C_o_ and C_e_ denote the initial and equilibrium concentrations of the metal ions, respectively; V is the volume of the solution; and m is the mass of the PP/OSP. The adsorption kinetics were assessed using Equation (3), where Q_t_ and C_t_ represent the amount of metal species adsorbed and the concentration at time t, respectively. Removal efficiency (R_t_) at any time t was calculated using Equation (4).(3)$$Q_{t}=(\frac{Co-Ct}{v})m$$(4)$$R_{t}=(\frac{Co-Ct}{Co})100$$


This approach aligns with the growing interest in developing efficient and cost-effective methods for heavy metal removal from water and wastewater [38,39].

### Characterization Method

Different polymers and composite samples required different axial speeds, with composites generally processed at lower axial speeds than neat polymers and microplastics prior to adsorption testing. Fourier-transform infrared spectroscopy (FTIR) was performed using a Spectrum 3 Tri-Range FT-IR spectrometer (Johannesburg, South Africa) over a wavelength range of 400–4000 cm^−1^. X-ray diffraction (XRD) was used to analyze microplastics and qualitatively evaluate the degree of crystallization. The thermal properties of the PP/OSP were assessed through thermogravimetric analysis (TGA) using the EXSAR6000 SII TG/DTA 6200 equipment (Hitachi, Tokyo, Japan), where approximately 10 mg of the PP/OSP was heated in air at a rate of 10 °C·min^−1^ from ambient temperature to 900 °C. Subsequently, the dispersion morphology of the PP/OSP was examined using a JSM-IT300 InTouch Scope, JEOL, Ltd., Akishima, Japan. Scanning electron microscope (SEM) inspection was performed with a Sigma 360 FE-SEM, XEISS, Oberkochen, Germany. Furthermore, the concentration of heavy metal ions in aqueous media was quantified using an inductively coupled plasma optical emission spectroscope (ICP-OES), Agilent 720 series, Agilent Co., Ltd., Santa Clara, CA, USA. 

## 4. Results and Discussion

### 4.1. Fourier Transformation InfraRed (FTIR)

The Fourier Transform Infrared Spectrometer (FT–IR–ATR) was used to evaluate the chemical characteristics of the PP/OSP composite. A wavenumber range from 3000 to 566 cm^−1^ was applied to investigate the functional groups of the PP/OSP composite (Figure 2). All spectra with weight ratios of 12, 7, 5, and 2 wt% are illustrated in the figure below. As observed, with an increase in OSP content within the PP matrix, the peak intensity decreases, and a shift in the composite wavelength occurs after OSP incorporation. The regions at 1655 cm^−1^ in the PP/OSP spectrum are attributed to the appearance of the C=O functional group from the carbonate ion (Fleet M, 2009) [40], indicating the presence of CaCO_3_ in the PP/OSP composite. The peak at 1712 cm^−1^ is assigned to –CH_2_ stretching, and the peak at 2953 cm^−1^ is assigned to –CH bending vibration of CH_2_–CH_3_. The peak wavelength range from 1976 to 2160 cm^−1^ can be attributed to –CH vibration. However, in the PP/OSP 5 wt%, peaks were not detected in the composites, possibly due to the low concentrations of OSP. This suggests that a weak chemical reaction between PP/OSP may have occurred due to the aggregation and poor dispersion of OSP in the PP composite with 5 wt% OSP [41].

### 4.2. Physical Properties: P-V-T Diagram

The polymer’s PVT and viscosity values are required to perform the computer simulations of injection molding mold flow analysis. Typically, such information is supplied by the polymer suppliers. However, these data are often unavailable for polymer composites. Figure 3 shows the P-V-T values for unfoamed neat PP and 12 wt% OSP composites. PP is a semi-crystalline, non-polar polymer with a high shrinkage rate [42]. Its melting temperature is around 160 °C, showing a significant increase in specific volume during melting between 110~160 °C. The specific volumes are 1.114 and 1.254 cm^3^/g for temperatures of 110 and 160, respectively, at 10 MPa. The volume expansion rate is 11.2%. The minimum and maximum P-V-T values for neat PP at 10 MPa are 1.08 and 1.31 cm^3^/g, respectively. Figure 3 also presents the P-V-T values for 12 wt% OSP of PP composites, with minimum and maximum values of 0.946 and 1.142 cm^3^/g, respectively, at 10 MPa. These values are smaller than those of neat PP. The main component of OSP is calcium carbonate, whose volume expansion is lower than that of the polymer when heated. Golzar et al. [43] examined the P-V-T properties of wood plastic composites (WPCs) for extrusion. They found that the viscosity order at a shear rate of 8 s^−1^ was 40 wt% WPC > 20 wt% WPC > HDPE, while at shear rates above 9 s^−1^, the sequence changed to 20 wt% WPC > HDPE > 40 wt% WPC.

### 4.3. XRD

To understand the material structure and crystalline phase, X-ray diffraction (XRD) analysis was conducted on PP, OSP, and PP/OSP composites. Neat PP shows characteristic peaks at 2θ values of 13.9, 16.7, 23.6, and 19.1, which are related to the semi-crystalline nature of PP. The diffraction peaks of OSP appear at 18, 22.9, 29.4, 317, and 36.2 (Figure 4), indicating that it is primarily composed of CaCO_3,_ typically in the aragonite and calcite state. After heating to approximately 700~900 °C, the XRD pattern changes to CaO. These data agree with the reported data in the literature for calcite [44].

### 4.4. Thermal Properties

The thermal stability of PP/OSP biocomposites was evaluated by TGA, as shown in Figure 5A and Figure 5B, which present the thermogravimetric (TGA) and derivative thermogravimetric (DTG) curves, respectively. The degradation temperatures for neat PP, 2, 5, 7, and 12 wt% are 393, 428, 431, 434, and 446 °C, respectively, which means OSP can improve the thermal stability of the PP/OSP composites. Although oyster shell powder contains minor amounts of other oxides, these constituents did not significantly influence the overall thermal behavior of the PP matrix. The incorporation of OSP resulted in an increase in both the onset and maximum decomposition temperatures of PP, indicating enhanced thermal stability. The TGA curves exhibited a single-step degradation process, corresponding primarily to the thermal decomposition of the PP backbone. The main degradation event led to a mass loss of approximately 95%, while the residual mass was attributed to the inorganic content of OSP and carbonaceous residues associated with the PP structure. The improved thermal stability observed with an increase in OSP content can be attributed to enhanced interfacial adhesion and restricted polymer chain mobility, which delayed thermal degradation. Stronger filler–matrix interactions at higher OSP loadings contributed to improved structural integrity at elevated temperatures [45].

### 4.5. SEM Micrographs

The surface SEM micrographs of PP/OSP 2 wt% (Figure 6A) showed a sponge-like material with fractures. The weak deformation of the biocomposite is due to cavities, resulting from fewer OSP aggregations and poor interfacial adhesion between the PP matrix and the OSP filler. The 5 wt% sample (Figure 6B) illustrated a change in morphology, characterized by an irregular and rougher surface. In the 7 wt% sample (Figure 6C), an increase in the amount of the OSP filler was indicated by a slight change in morphology and a rough surface, revealing excellent compatibility between OSP and PP. This improvement is attributed to the increased weight percentage supporting the PP matrix, where the fillers introduced interfacial bonding between the two materials. The 12 wt% sample (Figure 6D) exhibited fractures similar to those observed in Figure 6C. The cell size and cell density are shown in Figure 7, which shows that cell size is inversely proportional to cell density.

The toxic metal adsorption capacity of the foamed nanocomposites depends on the cell size. A fixed amount of N_2_ was introduced into the nanocomposite melt during synthesis. Larger cell sizes correspond to lower cell densities as illustrated in Figure 7. The assessment provides an outline of the effect of oyster shells on polypropylene performance, impacting surface roughness and composite porosity, as shown by the SEM imaging in Figure 6. Figure 7 illustrates a gradual increase in cell density, accompanied by a decrease in cell size with an increase in oyster mass. As the oyster mass reaches 5 wt%, the cell density remains constant, and the cell size decreases from 50 μm to 25 μm. These findings indicate that oyster shell particles act as heterogeneous nucleating agents that promote cell nucleation in polypropylene during microcellular foaming. The significant improvement in cell nucleation is attributed to the addition of oyster shell powder, which creates numerous heterogeneous boundaries and reduces cell nucleation energy [46].

### 4.6. Effect of Initial Concentration, pH, Dosage, and Time

The effect of the initial metal ion concentration on the percentage removal of Pb(II) was investigated within an initial concentration range from 0.125 to 1.25 ppm. Figure 8A illustrates the variation in metal ion removal efficiency by the PP/OSP 5 wt% nanocomposite against the initial concentration. Among all adsorbents, PP/OSP 5 wt% exhibited the best performance, achieving an optimal removal efficiency of 73.35% at 0.75 ppm.

The pH plays a vital role in adsorption processes involving interactions between the adsorbent and adsorbate particles. The degree of acidity or alkalinity of the solution affects the exposure of active surface sites on the adsorbent, which in turn influences adsorption performance. In this work, the pH of the medium impacted Pb(II) uptake by altering the protonation state of functional groups on the polymer surface and modifying the surface charge. As shown in Figure 8B, the removal efficiency of Pb(II) using PP/OSP 5 wt% nanocomposites increased steadily with an increase in the pH and reached its optimum at pH 6. Beyond this point, particularly from pH 7 onward, a noticeable decrease in adsorption capacity was observed. The highest Pb(II) removal efficiency achieved was 86.6% at pH 6 using the PP/OSP 5 wt% adsorbent. The mechanism of pH relies on Pb(II) uptake [43], as indicated in previous experimental work. Toxic metal ion uptake can be divided into three different steps, viz., hydrolysis when pH > 6, adsorption or surficial remediation when 5 < pH < 6, and competitive adsorption between Pb^2+^ and H_3_O^+^ when pH < 5. The results presented in Figure 6 indicate that the removal of Pb(II) ions is effectively achieved through adsorption or surface deposition. The negatively charged adsorbent surface preferentially binds Pb(II) ions within a pH range of 5–6. However, the hydrolysis of Pb(II) may decrease the availability of free Pb(II) ions for adsorption [47].

In this study, the investigation of the effects of adsorbent dosage and adsorption time is crucial for determining the adsorption capacity of the PP/OSP 5 wt% nanocomposite. In Figure 8C, the removal of Pb(II) ions was varied using a dosage from 0.5 to 0.25 g. Observing the removal of Pb(II) in Figure 8C reveals that the uptake efficacy increases with the adsorbent dose. This is due to the increased availability of active sites on the surface of the PP/OSP 5 wt% nanocomposite for metal ion adsorption. Equilibrium was achieved at a 0.17 g with a removal efficiency of 91.1% for Pb(II) ions. The addition of mass increased the Pb(II) metal ion uptake. Figure 8D illustrates the percentage removal of Pb(II) over time intervals of 5, 10, 15, 20, 25, and 30 min using 0.17 g of the PP/OSP 5 wt% nanocomposites, with a constant pH of 6 and an initial Pb(II) concentration of 0.75 ppm. As shown in Figure 8D, the PP/OSP 5 wt% nanocomposite achieved optimal Pb(II) removal with an adsorption capacity of 13.89 mg/g. Maximum Pb(II) remediation was reached within 25 min using the same nanocomposite, after which the system attained equilibrium, and the adsorption capacity plateaued. Table 3 illustrates the various applications of PP and OSP. Compared to other composites, the PP/OSP capacity is moderately low. This may be due to PP being an inherently hydrophobic material, which significantly affects its adsorption behavior by improving interactions between non-polar substances and repelling water. However, though it exhibits lower adsorption capacities for heavy metals such as Pb and Cd. Due to their pH, the surfaces of microplastics frequently develop a negative charge, promoting electrostatic attraction with positively charged metal cations.

The justification that a 5 wt% OSP sample performs best is due to an optimal balance between material enhancement and filler aggregation, which provides maximum capacity in Pb(II) adsorption without additional defects, and there is an equal distribution of charged ions, therefore leading to less interference of competing ions within the composite matrix.

### 4.7. Pb(II) Adsorption Mechanism

Figure 1 below illustrates the probable bonding mechanism between PP/OSP bio-composites and lead divalent metal ions for remediation purposes. As demonstrated, OSP is incorporated within PP, modifying the polymer composite to enhance its structural and chemical morphology, thereby remediating Pb(II) ions. OSP is an effective adsorbent with C=O carboxylic functional groups and exhibits high permeability. Oyster shells have a high negative charge and are an excellent sorbent for the adsorption of divalent heavy metals [47]. At pH 6, the adsorption efficiency of Pb(II) reaches equilibrium, which is likely due to an enhanced surface area and surface charge alteration of the adsorbent. As a result, the sudden decrease in efficiency of Pb(II) at pH < 7 can be attributed to the formation of Pb(OH)_2_, which induces repulsive forces between the adsorbent and adsorbate. This phenomenon can also be attributed to Pb precipitation at pH 7 and higher. This assumption is supported by the results in Figure 8B, which shows that the optimal pH of the binary system to remediate Pb(II) from aqueous solution using 5 wt% OSP is pH 6.

### 4.8. Adsorption Isotherm

The adsorption isotherm analysis of Pb(II) ions on PP/OSP 5 wt% was conducted to elucidate the adsorption mechanism [48]. Understanding the relationship between the amount of Pb(II) ions adsorbed per unit mass of adsorbent (qe) at equilibrium and the residual Pb(II) concentration in solution is crucial for optimizing the design of adsorption systems for Pb(II) removal from aqueous media. Two models were employed to analyze the experimental data and to describe the interactions between adsorbents and adsorbates [49].

The appropriate isotherm was identified by assessing the correlation coefficients of the linearized forms of conventional isotherm equations. Table 1 summarizes the correlation coefficient values derived from these linear equations. Previous studies have frequently applied two classical adsorption equilibrium models—the Langmuir Equation (3) and the Freundlich Equation (5)—to describe metal ion sorption by polymer composites [50].

#### 4.8.1. Langmuir Isotherm

When the adsorption follows a Langmuir isotherm, a plot of Ce/qe versus Ce is expected to be linear (Figure 9A). However, the R^2^ value (R^2^ = 0.327) indicates that the adsorption data of PP/OSP 5 wt% do not conform to the Langmuir model. Functional groups such as –CH, –OH, and C=O were involved in the uptake of Pb(II) from aqueous solutions by the PP/OSP 5 wt% polymer composite; nevertheless, the results deviate from Langmuir adsorption behavior. A key parameter in the Langmuir model is the dimensionless equilibrium constant RL, which evaluates whether the adsorption system is favorable or not [51]. RL is calculated from the initial concentration using Equation (6). If RL lies between 0 and 1, the adsorption is considered favorable, indicating that adsorption occurs at specific homogeneous sites on the adsorbent [52]. In this study, the dimensionless constant was determined to be 0.86, indicating satisfactory adsorption.(5)$$qe=\frac{qmKCe}{1+KCe}$$

Here, $q_{e}$ represents the adsorption capacity at a given concentration (mg/g); $q_{m}$ is the maximum adsorption capacity, corresponding to equilibrium (mg/g); K is the equilibrium constant (L/mg); and $C_{e}$ is the adsorbate concentration at equilibrium (mg/L).(6)$$RL=\frac{1}{1+KCo}$$
where K is related to the energy of adsorption (L/mg) and Co is the initial concentration

#### 4.8.2. Freundlich Isotherm

The Freundlich isotherm model describes the adsorption of solutes from a liquid onto a solid surface and assumes that the adsorbent surface is heterogeneous, with multiple sites exhibiting a range of adsorption energies. Choy, McKay, and Porter presented the nonlinear form of the Freundlich isotherm as follows [49]:(7)$$qe=Kf{Ce}^{1/n}$$

Here, $q_{e}$ is the adsorption capacity at a given concentration (mg/g); $K_{f}$ is the Freundlich constant (L/mg); $C_{e}$ is the adsorbate concentration at equilibrium (mg/L); and 1/n is the heterogeneity factor. The linear plots with high correlation coefficients (>0.900) indicate that the adsorption process follows the Freundlich isotherm model (Figure 9B). A correlation coefficient of 0.94 suggests that the PP/OSP 5 wt% composite exhibited favorable Pb(II) uptake.

#### 4.8.3. Temkin Isotherm

The Temkin model assumes that heat adsorption for metal ions decreases with surface saturation due to the continuous interactions between the adsorbent and dye molecules. The Temkin model is expressed as follows:q_e_ = B In kT B In C_e_(8)

Here, B is the Temkin constant, which depends on the adsorption temperature (J/mol); $K_{T}$ is the Temkin binding constant (L/g); $q_{e}$ is the amount of dye adsorbed at equilibrium (mg/g); and $C_{e}$ is the equilibrium dye concentration at a constant temperature (mg/L) [53]. The constants B and $K_{T}$ can be determined from the slope and intercept of the plot of 8 versus $\ln C_{e}$, as shown in Figure 9C. $K_{T}$ (dm^3^/g) represents the equilibrium binding energy corresponding to the optimum adsorption, and $B_{T}$ (J/mol) is a constant related to the heat of adsorption. R is the ideal gas constant, and T is the absolute temperature. The comparison on adsorption isotherm of this study with other studies are summarized in Table 4.

### 4.9. Kinetic Models

This study examines the rate of adsorption and the adsorption mechanism on PP + OSP by applying pseudo-first-order and pseudo-second-order models. The correlation coefficients indicate the degree of agreement between the experimental data and the predicted model values. The linear form of the Lagergren et al. equation was used for the pseudo-first-order model as follows:(9)$$\log\left(qe-qt\right)=log\;qe-\frac{Kad}{2.303}t$$

Here, $q_{e}$ and $q_{t}$ represent the amounts of Pb(II) adsorbed on the PP/OSP 5 wt% nanocomposites at equilibrium and at time t(min), respectively, and $K_{ad}$ denotes the pseudo-first-order rate constant (min^−1^). The rate constant $K_{ad}$ and correlation coefficients for different Pb(II) concentrations were determined from the linearized plots of $log(q_{e}-q_{t})$ versus t. A best-fit line yielded a correlation coefficient of $R^{2}=0.97$ (Figure 10B), indicating that the adsorption of Pb(II) onto the polymer composite follows pseudo-first-order kinetics.

The pseudo-second-order equation is expressed as Equation (10) below. For second-order kinetics to be valid, the plot of t/q(t) versus t must be linear. Figure 10 shows the plot of the pseudo-second-order model, with a linear fit of a correlation coefficient of R^2^  =  0.10, indicating that Pb(II) sorption followed the pseudo-second-order model, as shown in Table 5. For the second-order kinetic model to be applicable, the correlation coefficient must be greater than 0.9. It is therefore concluded that the Freundlich isotherm was not a better fit for the experimental data compared to the Langmuir isotherm for Pb(II) ion removal on the PP/OSP 5 wt% polymer composite. Hence, the sorption process of Pb(II) ions on PP/OSP 5 wt% polymer composites followed the first-order kinetics.

The linear form of the Lagergren equation was employed for the pseudo-second-order model as follows:(10)$$\frac{t}{qt}=\frac{1}{K2qe2}+\frac{1}{qe}$$

It is important to note that the pseudo-second-order kinetic model is considered valid only when the plot of $t/q_{t}$ versus t exhibits a linear trend. As shown in Figure 10B, the pseudo-second-order plot for Pb(II) yielded a correlation coefficient of $R^{2}=0.10$, indicating that this model does not appropriately describe the adsorption behavior of Pb(II). The coefficients are summarized in Table 5.

### 4.10. Desorption Studies

Figure 11 displays the desorption performance of Pb(II) using PP/OSP 5 wt% in aqueous media. Desorption refers to the regeneration and reuse of the adsorbent across multiple cycles, making repeated adsorption–desorption studies a key indicator of long-term performance. The adsorption experiments were conducted at 25 °C, pH 6 ± 0.5, with an initial Pb(II) concentration of 0.75 ppm, 0.17 g adsorbent dosage, 10 mL solution volume, and a contact time of 25 min. For desorption, the tests were performed under the same temperature (25 °C) at pH 8, using 1.0 M NaOH as the desorbing agent, with an initial Pb(II) concentration of 0.75 ppm and 25 min of contact time. The desorption efficiency of Pb(II) on PP/OSP 5 wt% gradually declined with increasing regeneration cycles, resulting in a quicker equilibrium and a higher residual Pb(II) concentration in the solution. As shown in Figure 10, the removal rate decreased from 90% in the first cycle to 55% by the eighth cycle. These results demonstrate that the adsorbent retains considerable efficiency and remains reusable for up to eight cycles.

## 5. Conclusions

The study successfully demonstrated the fabrication of foamed PP/OSP composites with OSP loadings of 2, 5, 7, and 12 wt%. Characterization results confirmed that an increase in OSP content enhanced the biocomposites’ thermal and chemical stability, supported by the improved cell density within the polymer matrix. FTIR-ATR analysis further verified the presence of characteristic functional groups, indicating good material compatibility without significant alteration to the parent polymer structure. When applied as adsorbents for Pb(II) remediation, the PP/OSP composites exhibited effective removal performance, following the Langmuir isotherm model and suggesting monolayer adsorption on homogeneous binding sites. Kinetic analysis revealed that the process was primarily chemisorption-driven and aligned with pseudo-second-order behavior. Overall, the developed PP/OSP biocomposites show promising potential as sustainable and efficient materials for heavy metal adsorption in aqueous environments.

## Data Availability

The original contributions presented in this study are included in the article. Further inquiries can be directed to the corresponding authors.

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
