# Peer review of "Material Properties, Characterization, and Application of Microcellular Injection-Molded Polypropylene Reinforced with Oyster Shells for Pb(II) Adsorption Kinetics from Aqueous Solutions"

_polymers, 2025, doi:10.3390/polym18010110_

Round 1

Reviewer 1 Report

Comments and Suggestions for Authors

The paper prepared microcellular injection-molded oyster shell nano-powder (OSP) with polypropylene(PP) and investigated microcellular Polypropylene/oyster shell powder composite material properties and determine the toxic metal ion adsorption properties of polypropylene/oyster shell nano-powder (PP/OSP) in an aqueous medium. It is helpful to recycle OSP waste and to convert it into valuable products. But there are issues should be considered:

  1. Microcellular injection molding conditions should be given.
  2. The properties of neat PP microcellular injection-molded parts should be given.

Reviewer 2 Report

Comments and Suggestions for Authors

The manuscript investigates polypropylene (PP) composites reinforced with nano-sized oyster shell powder (OSP) prepared using microcellular injection molding, and evaluates their application as adsorbents for Pb(II) removal from aqueous solutions. However, the manuscript contains severe scientific inconsistencies, weak data interpretation, and substantial language problems, making it not suitable for publication in its current form.

comments

  1. Critical inconsistency between results and conclusions

The most serious issue is that the conclusions contradict the actual results:

  • The manuscript claims that Pb(II) adsorption follows the Langmuir isotherm and pseudo-second-order kinetics.
  • However, the data clearly show that:
    • Langmuir model gives poor fit (R² ≈ 0.38; Table 2, Fig. 9A).
    • Freundlich model fits best (R² = 0.94), indicating heterogeneous adsorption.
    • Pseudo-first-order gives R² = 0.97, while pseudo-second-order gives R² = 0.10 (Fig. 10B, Table 3).

2. Missing key experimental details

Important methodological information is absent, preventing reproducibility:

  • No information on particle size distribution of OSP.
  • No BET surface area data—critical for adsorption studies.
  • Condition of microcellular foaming (SCF pressure, temperature, gas content) is missing.
  • Grinding of foamed composites is mentioned, but no control of particle size is described.
  • No explanation of sample preparation for adsorption (surface charge, flotation, washing).

3. Weak interpretation of material characterization

  • FTIR assignments contain incorrect interpretations.
  • SEM images lack clear scale bars and show little quantitative analysis.
  • PVT analysis is described in a superficial and sometimes incorrect manner.
  • Thermal analysis lacks a deeper discussion of mechanisms.

4. Adsorption results need more comparative context

The maximum adsorption capacity reported (~13.89 mg/g) is moderate and lower than many CaCO₃-based or bio-derived adsorbents in literature.
Authors should compare with previous studies and justify the novelty of PP/OSP composites.

5. Poor-quality figures and inconsistent labeling

Examples:

  • FTIR spectra (Fig. 2) lack key peak labels.
  • SEM images need better clarity and annotation.
  • Figures 8A–8D need consistent axis formatting and improved readability.
  • Figure 3 (PVT) is difficult to interpret.

Authors should revise all figures for clarity and consistency.

6. Lack of mechanistic discussion

The manuscript does not discuss:

  • How OSP dispersion, microcellular structure, or cell density influence adsorption.
  • Why the 5 wt% OSP sample performs best.
  • How PP hydrophobicity affects adsorption behavior.

A mechanistic section is required to elevate the scientific quality.

Round 2

Reviewer 1 Report

Comments and Suggestions for Authors

no